**Subject Category:**
Biology (whole organism)

environmental science

conservation costs, threatened species recovery, Lear's macaw, rewards

**Author for correspondence:**
Antonio E. A. Barbosa
e-mail: antonio-eduardo.barbosa@icmbio.gov.br

# How much does it cost to save a species from extinction? Costs and rewards of conserving the Lear's macaw

## Antonio E. A. Barbosa[1] and José L. Tella[2]

[1]The National Center for Bird Conservation and Research (CEMAVE), BR 230, Floresta Nacional da Restinga de Cabedelo, Renascer, 58.108-012 Cebedelo-PB, Brazil
[2]Department of Conservation Biology, Estación Biológica de Doñana (CSIC), Avenida Américo Vespucio s/n, 41092 Sevilla, Spain

 AEAB, 0000-0001-9921-4479; JLT, 0000-0002-3038-7424

Although the limited resources available to save species from extinction necessitate the optimization of conservation actions, little is known about their costs and effectiveness. We developed a costs–rewards framework that integrates information on which sectors of society contribute to funding conservation, how much is contributed, how funds are distributed among conservation targets and how these investments drive not only conservation rewards but also the economic and ecosystem services that benefit society. We applied this framework to the Lear's macaw (*Anodorhynchus leari*), a species discovered in the wild in 1978 with only 60 individuals. Funds invested over the last 25 years reached US$3.66 million. The contribution of governments, non-governmental organizations and private funders varied over time, as did the funding targets. Funds were proportionally invested to mitigate the main causes of mortality, while no funds were devoted to protecting foraging habitats. Conservation rewards were satisfactory, with the cost and time needed to downlist the species from critically endangered to endangered being similar to those invested in other bird species. However, economic rewards (through ecotourism and handicrafts linked to the conservation of the species) were low and require promotion, while ecosystem services provided by Lear's macaws have yet to be quantified.

## 1. Introduction

Governments, private sectors and civil society invested approximately US$21.5bn between 2001 and 2008 into global

**Figure 1.** The Lear's macaw was discovered in the wild in 1978, with a population estimated at only 60 individuals.

conservation efforts [1]. However, we have failed to reduce the rate of biodiversity loss despite this substantial investment [2]. Both the inadequate levels [3] and the low cost-effectiveness ratio of funds invested are noted as major impediments to achieving conservation goals [4]. A recent study estimated that there is less than 15% of the funds required to downgrade the conservation status of all currently threatened bird species, as identified by the International Union for Conservation of Nature's (IUCN) Red List of Threatened Species, over the next decade [5]. Therefore, incorporating management costs into conservation planning to maximize results should be the golden rule, since conservation resources are always limited [6].

Despite the increasing concern about the effectiveness of conservation interventions and the extensive literature highlighting the importance of integrating an economic approach into conservation efforts over the last several years [6–8], there have been few attempts to assess cost-effectiveness of conservation actions. Costs have not been considered basically because of the difficulty in obtaining economic data [9] and the fact that most biologists are not familiar with economic concepts such as cost-effectiveness, or with collecting relevant economic data [6]. Therefore, there are few studies addressing the costs of conserving habitats [10–12] and communities [9,13,14], with even less attention paid to the management of single species [15–17]. Consequently, our predictive capacity to identify cost-efficient priorities is hampered by the scarcity of data on actual conservation spending, particularly at a fine scale [10]. On the other hand, there are few assessments of the rewards of conservation actions beyond the conserving of target species, such as economic rewards gained by local communities or the restoration of ecosystem services [18–20].

Conservationists often focus on *in situ* management actions aimed at saving species from extinction. This is true for the Lear's macaw (*Anodorhynchus leari*, figure 1), after its discovery in the wild in 1978 in northeastern Bahia state, Brazil, with a global population estimated at only 60 individuals [21,22]. Evidence gathered in the field on considerable human pressures and the vulnerability of this small population [22,23] contributed to its inclusion on the IUCN's Red List as Threatened in 1988 and as critically endangered (CR) in 1994 [24]. This alarming scenario prompted the development of the first conservation initiatives. One of these initiatives was organized in 1993 by the Brazilian government, which established a group of specialists that would be responsible for outlining strategies aimed at the recovery of the Lear's macaw. Thereafter, several monitoring and research projects aimed at conservation were promoted within the prioritized alignments later compiled by the first official management plan (published in 2006) and by the second national action plan for the conservation of the Lear's macaw [25]. Such efforts appeared to be fruitful as the species has recovered in numbers over the past few decades, allowing its downlisting from CR to endangered (EN) by the IUCN in 2009 [24]. However, there is no compiled information on the overall costs of the different conservation projects, and it is not clear to what extent the different conservation actions contributed to the recovery of the species, given the absence of assessments of their costs and effectiveness. Likewise, information is scarce about economic rewards [26] and is non-existent for the potential ecosystem services recovered.

Here, we aimed to compile detailed information on the costs and rewards of the *in situ* conservation of the Lear's macaw. We built a novel costs–rewards framework that integrates detailed information on which sectors of society contributed to funding the conservation, how much was contributed, how funds were

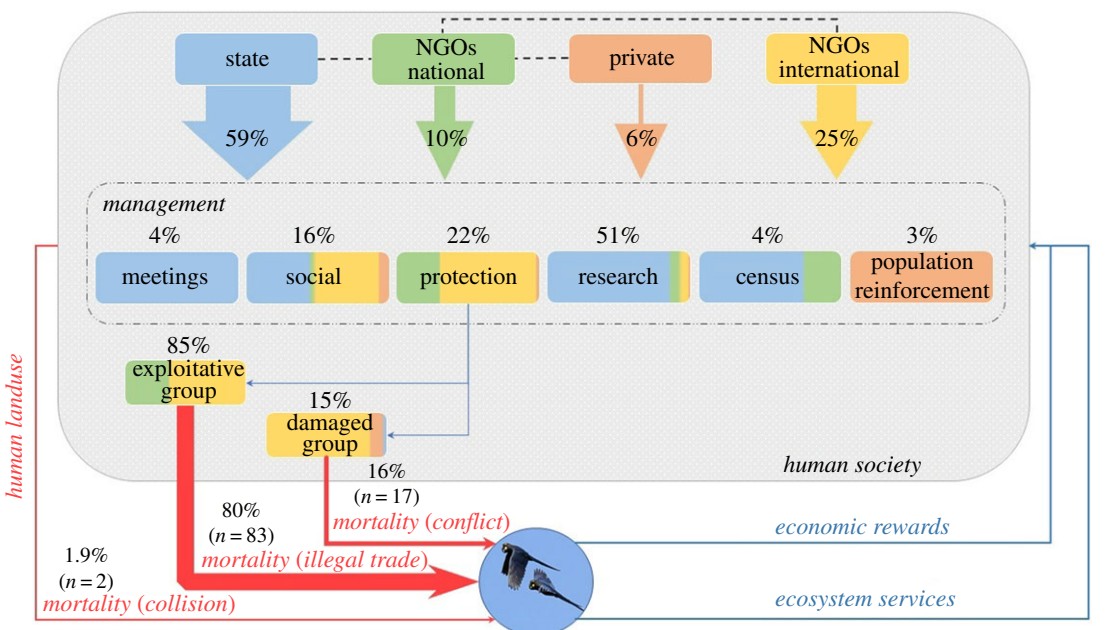

**Figure 2.** Flux of different funding sectors to different conservation targets and the main threats affecting the conservation of the Lear's macaws, as well as potential economic and ecosystem services rewards. The width of the red arrows is proportional to the main reported causes of death and extraction of macaws from the population. The colours in the boxes represent the proportions of funds from each funding sector to different conservation targets: state (blue), national NGOs (green), private (orange) and international NGOs (yellow).

distributed among different conservation targets, including actions aimed to mitigate impacts caused by the exploitative (poachers) and damaged (farmers) sectors of society (causing poaching and killing of macaws, respectively) as well as other human-caused impacts, and how investments in conservation drove not only conservation rewards but also economic and ecosystem services (figure 2). Specifically, our goals within this framework were to estimate (i) the total economic costs, (ii) the temporal trends since the discovery of the species, (iii) the contribution of different funding sectors (state, private, national and international non-governmental organizations (NGOs)), and (iv) the overall and temporal conservation targets of these funds. We also assessed whether (v) funds were proportionally targeted to mitigate the different conservation threats to the species. For this purpose, we tried to estimate the relative contribution of habitat loss and human-caused sources of mortality and extraction of individuals from the population. Finally, we attempted to assess rewards in terms of (vi) mitigation of conservation threats and the recovery of the population and (vii) economic rewards provided by Lear's macaws.

# 2. Material and methods

## 2.1. Study species

The Lear's macaw *A. leari* is endemic to the arid 'Caatinga' biome. Its distribution is restricted to a small area in northeastern Bahia state, Brazil, and is mainly concentrated in two currently protected areas separated by 38 km, where they breed and roost communally in large sandstones cliffs, called Raso da Catarina Ecological Station (RCES) and Canudos Biological Station (CBS) [25]. The birds perform daily movements from these sites to forage in neighbouring unprotected areas within a radius of *ca* 70 km [27–29], including eight municipalities (Jeremoabo, Canudos, Euclides da Cunha, Santa Brígida, Paulo Afonso, Glória, Rodelas, Monte Santo). Recently, the population has slightly expanded by recolonizing historical breeding sites within its foraging areas [30]. There is also a marginal and isolated population composed of two individuals located 135 km to the west, in the municipalities of Sento Sé and Campo Formoso [25,31]. A large proportion of the population (80%) is composed of non-breeding individuals [32].

The most well-known conservation threats to the species are the agropastoral activities contributing to the loss and degradation of the habitat, especially those affecting an important food resource for the species, namely the fruits of the licuri palm (*Syagrus coronata*). More directly, poaching for illegal trade

and killing by farmers, because of corn crop damage by macaws, cause the removal of individuals from the wild [25]. Hunting macaws for food also apparently placed strong pressures on populations in the past (E. Pacifico 2018, unpublished data).

## 2.2. Funds invested

Information on funds invested was obtained through personal communications with project officers, managers and employees from all institutions that have funded, executed or supported Lear's macaw conservation actions. Information was requested for each project/activity expenditure for which the resource was used, as well as its source of funding, to avoid duplication of information. Unfortunately, it was not possible to contact the leadership of two NGOs (BioBrasil Foundation and PROAVES), as they were closed down several years ago. PROAVES and the BioBrasil Foundation conducted conservation work during 2001–2007 and 1995–2006, respectively. Nonetheless, we were able to obtain records of expenditure by both NGOs, which were available in the form of internal documents from the National Center for Bird Conservation and Research (CEMAVE/ICMBio), information provided to us by ex-employees as well as through the institutions from which the funding resources were obtained. Some data were also collected from electronic magazines and published and unpublished reports from CEMAVE/ICMBio.

In addition, the leadership of the Biodiversitas Foundation did not respond to our requests for detailed information on expenditure. Despite this, we decided to include the Biodiversitas Foundation in our study using estimates, since this NGO provides sizeable investments for the CBS to protect the sandstone cliffs where a significant fraction of the world's population of Lear's macaws breed and roost communally. These investments included costs such as the wages of park rangers and a biologist, and land acquisition and infrastructure that were exclusively supported by this NGO. This information was obtained in conversation with a park ranger who had worked at this location since the creation of the reserve and through consulting the institutions that funded the projects led by this NGO. The wage payment costs were conservatively estimated by considering the minimum wage in Brazil following its annual readjustment (available at Portal Brazil [34]). We opted to be more conservative by not considering mandatory expenses, such as federal fees or any bonuses paid to employees, as they can vary for the same position depending on the level of education or responsibilities. Thus, this would lead to an overestimation of the amount invested by the Biodiversitas Foundation.

Regarding information from the Brazilian government, the ICMBio does not maintain a public database with the detailed costs of activities exclusive to the Lear's macaw. However, we relied on the help of the ICMBio central stewardship to obtain detailed information on expenditure related to the Lear's macaw between 2010 and 2017, and on the construction and maintenance expenditure of the research base (2004–2017) thanks to the Administration and Finance Unit (UAAF) of ICMBio. We did not include in our analysis the expenses related to office supplies and goods from the research base, such as vehicles, furniture and office equipment. First, we were not able to find any records of these expenses. Second, the research base's goods were later redirected to the institution's parent organization.

Information from previous periods was obtained by consulting internal documents, published and unpublished reports from CEMAVE/ICMBio, as well as the Rates Concession and Ticket System (SCDP) from the Federal government [35]. The information from the government employees' wages was obtained through personal communication.

Altogether, the information we were able to obtain from a variety of sources is surely an underestimation of the actual investment in the conservation of the Lear's macaw. However, we believe this underestimation is not strongly biased towards public or private funding sources.

## 2.3. Analysis of the expenditure

We analysed the spending involved in *in situ* conservation in its broader sense (including management, population monitoring, conservation-oriented research and social work with local communities) of the Lear's macaw between 1992 and 2017. We summed annual funds invested in each kind of conservation action to obtain total annual expenses. We also classified all investments based on the origin of the funding sectors: (i) public funds from the State, including those of the Federal and State government agencies, (ii) funds from international NGOs, (iii) funds from national NGOs, and (iv) funds from private sources, including commercial companies and individual funders such as aviculturists. Likewise, we classified all funds based on the different categories of conservation targets, defined as follows.

*Protection*: expenses related to the protection of the species and its habitat (e.g. management, surveillance of protected area) and the reduction in human-induced mortality of birds (i.e. corn reward program, rescue of injured birds and poaching; see below). We did not take into account RCES expenditure (i.e. employees' wages, creation and operation of the protected area), as this protected area was created before the discovery of the Lear's macaw [21] and activities there are not targeted to this species. Likewise, costs of surveillance operations by Federal and State agencies were not included for many reasons. First, there is no available systematized information about surveillance efforts in the area, much less its costs. Second, we do not have evidence of their effectiveness regarding this species, since, until now, there has been no record of Lear's macaw seizures in these operations carried out *in situ*; in all cases, the poached birds were seized outside of the small-range distribution of the species. Third, these operations were directed at various types of environmental crime. In any case, the proportional investment of RCES and surveillance operation expenditure for the Lear's macaw should represent an anecdotal amount of funds.

*Social*: expenses related to social activities involving local communities, such as environmental education, awareness and income generation projects (e.g. promoting the production of handicrafts related to the conservation of the Lear's macaw by local artisans).

*Research*: expenses invested in scientific research aimed at generating or updating information on the species to guide conservation actions. These investments contributed to the production of knowledge in different research areas, such as breeding biology [32,36], distribution and population dynamics [28,31,37], diet and foraging behaviour [29,38,39] and *in situ* management [40].

*Census*: expenses invested in the annual censuses of the entire population of the species.

*Population reinforcement*: investments in an ongoing project aimed at reinforcing the almost extinct population located in Boqueirão da Onça, northeastern Bahia, where two individuals have remained isolated from the main population (Raso da Catarina) since 1995 [25,41]. The costs considered here are exclusively from the implementation of the release project (i.e. facilities, training and monitoring of the birds).

*Meetings*: expenses related to holding meetings and workshops with the participation of the Brazilian government and experts on the species, with the purpose of planning and evaluating conservation initiatives.

All expenses measured in foreign currencies were converted, within the corresponding period, to Brazilian currency using the mean exchange rate of that year [33]. We should note that all foreign funding was supported by international institutions with annual project calls. Therefore, expenditure should be made within the year of funding concession. Additionally, using the citizens' calculator [42], we refreshed the estimated costs by applying the inflation correction index (IPCA-E IBGE), to determine its equivalence in current monetary terms (June 2017). Likewise, funds previous to the Brazilian Real currency were translated to the current currency.

All analyses were run in R 3.0.2 [43], and graphics were created in R 3.0.2 using ggplot2 [44] and Microsoft Excel (2016).

## 2.4. Conservation rewards

In an attempt to assess conservation rewards, we related the temporal trend in investments in conservation to the population trends of the species over the years, using data from annual censuses of the entire Lear's macaw population provided by Barbosa *et al.* [25] as mean annual counts. Moreover, we assessed the quantity of funds devoted to mitigate the main conservation threats to the species, and whether the funds invested were proportional to the potential impact of these threats as described below.

### 2.4.1. Habitat loss

We estimated changes in the extent of forest and agropastoral areas from the MAPBIOMAS Project web platform [45], which provides detailed annual information on land uses for the period 2000–2016. For this purpose, we selected the following municipalities that are included in the area of occurrence of the species, including foraging areas: Jeremoabo, Canudos, Euclides da Cunha, Santa Brígida, Paulo Afonso, Glória, Rodelas, Monte Santo, Sento Sé and Campo Formoso. The annual surfaces of forest and agropastoral areas from each municipality were summed to detect a possible recent trend in land use changes.

### 2.4.2. Removal of individuals from the wild

We used the information contained in the 2016 studbook of the species [46], as well as internal documents of CEMAVE/ICMBio, to estimate the yearly human-caused removal of wild birds. We

recorded the number of individuals that were seized (because of illegal poaching), that were killed by farmers or that died from other causes (e.g. collision with power lines).

## 2.5. Economic rewards

Currently, there are two activities of economic potential being developed in relation to the conservation of the Lear's macaw: ornithological tourism and handicrafts.

### 2.5.1. Ornithological tourism

We contacted the four local bird watching companies that offer guided visits to see the macaws at Canudos Biological Reserve, to determine the dynamics of visits and assess the potential economic rewards gained from ecotourism by the local community.

### 2.5.2. Handicrafts

We contacted the three handicraft associations linked to the conservation of the species, located within the species' area of occurrence, to obtain information on the number of workers from each association. After compiling information about the bulk sales from the handicrafts produced [47] and the mean family income from three communities [26], we also assessed the relative contribution from handicrafts to craftsmens' mean family income in three communities, linking their products to sustainable practices and Lear's macaw conservation. Although there are data available on the bulk sales from the Association of Artisans Lear in Serra Branca from the period 2011–2015 [47], we only used data from 2014 as a reference to estimate the relative contribution of handicrafts to average family income since it was the only year with information available from all three communities.

# 3. Results

## 3.1. Costs of conservation

### 3.1.1. Overall costs and temporal trends

Our compiled information reveals that at least R$7 096 143 (equivalent to approx. US$2 260 000) were spent in the last 25 years on *in situ* conservation of the Lear's macaw. This figure equals R$11 481 210 (approx. US$3 660 000) after adjusting for inflation to local currency. There was a sharp increase in the allocation of funds in the early 2000s, coinciding with the period in which the Federal government established a research base, followed by an apparent stabilization of investments in the years following the demobilization of permanent staff (figure 3a).

### 3.1.2. Sources of funding

The public sector contributed most of the resources (59%) invested in Lear's macaw conservation, accumulating R$4 156 588. International NGOs invested much more than Brazilian NGOs, accumulating R$1 764 962 (25%) and R$725 541 (10%), respectively. The private sector contributed only 6% of the investment (R$419 052) (figure 2).

There were large variations in the investments made by different supporters within each funding sector. Larger variations occurred among public institutions ($\pm$1 072 882 s.d.) and Brazilian NGOs ($\pm$179 919 s.d.), while there were lower variations within the private ($\pm$123 158 s.d.) and international NGOs ($\pm$175 070 s.d.)

The different funding sectors contributed differently over the conservation trajectory of the Lear's macaw. International and national NGOs sponsored the first small resources, followed by larger contributions from the public sector in the second half of the period and a recent contribution of private funds (figure 3b).

### 3.1.3. Funding targets

Most of the funds (51%) were devoted to research (R$3 571 029), followed by protection and social targets with 22% (R$1 531 178) and 16% (R$1 122 970) of the funds invested, respectively. The lowest investments

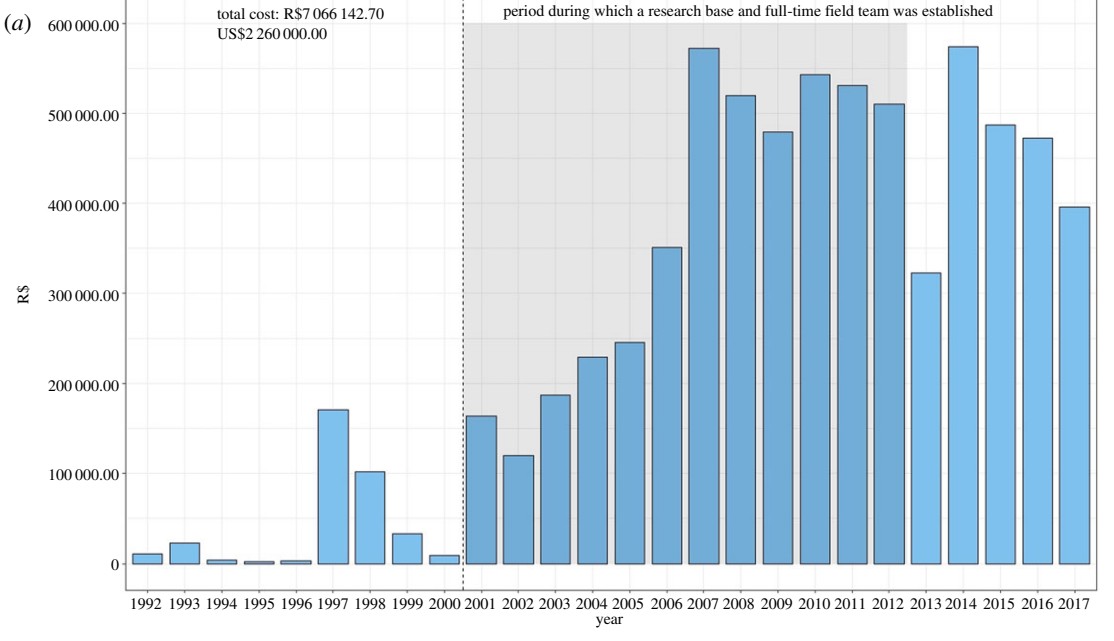

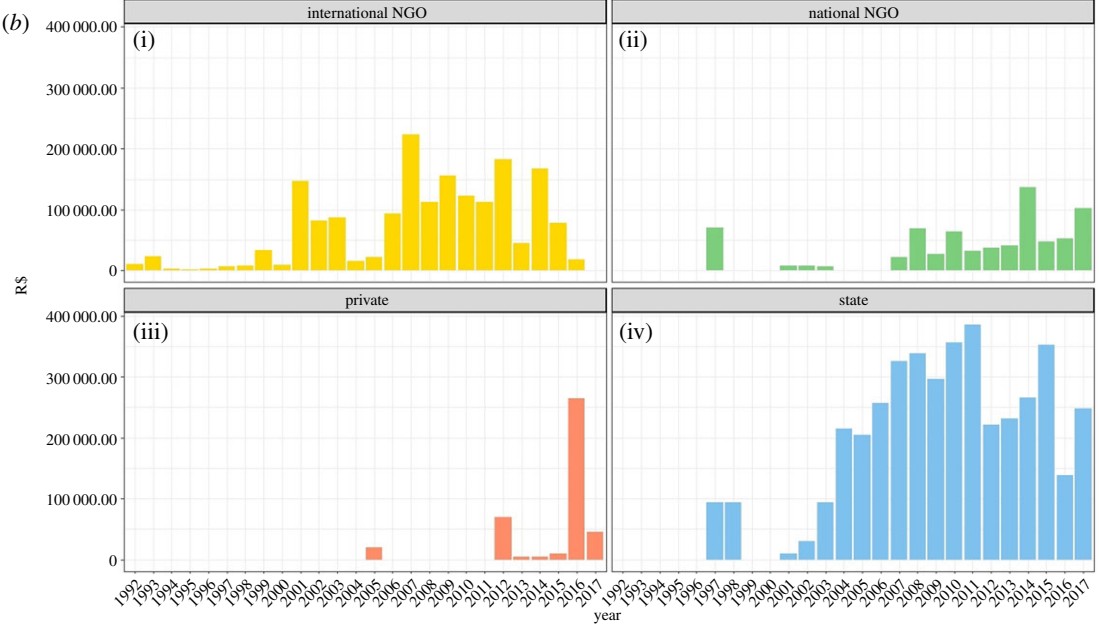

**Figure 3.** (*a*) Temporal changes in funds invested in the conservation of the Lear's macaw. (*b*) Temporal trends in funds invested by the different funding sectors.

corresponded to census (4%, R$298 763), meetings (4%, R$287 202) and population reinforcement (3%, R$255 000) (figure 2).

Different funding sectors contributed differently to each conservation target. In absolute terms (figure 2; see electronic supplementary material, figure S1), the public sector invested most of its funds to support research activities, comparatively little funds to censuses, meetings and social activities, and almost nothing to protection (figure 2; see electronic supplementary material, figure S1). Conversely, national and international NGOs were more likely to support initiatives aimed at protecting the species and its habitat. In addition, research and social activities were supported in different ways by these groups: while international NGOs were more likely to sponsor social activities than research, this trend was the opposite for national NGOs. The private sector contributed more evenly to different conservation targets, and was the only sector supporting population reinforcement (figure 2; see electronic supplementary material, figure S1).

The contribution of different funding sectors to each conservation target changed when examining their relative contributions (figure 2). The public sector supported most of the research but also all

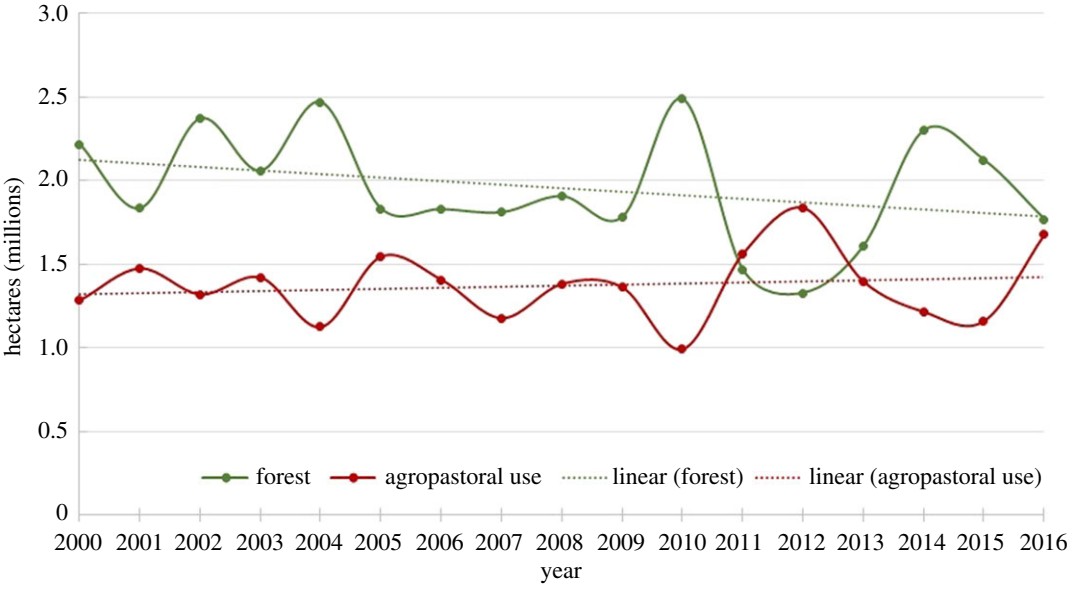

**Figure 4.** Temporal changes in the percentage of forest and agropastoral land uses in the 10 municipalities used by Lear's macaws as foraging areas.

meeting expenses and most of the census expenses; international NGOs supported most of the protection activities; social activities were equally supported by state and international NGOs; and the private sector was the sole sponsor of population reinforcement tasks (figure 2).

### 3.1.4. Funding invested in protection

Funds devoted to direct and indirect protection of the species represented 22% (R$1 531 178) of the total investment (figure 2).

#### 3.1.4.1. Assessing conservation threats

*Habitat loss.* Data obtained from MAPBIOMAS indicate a trend of forest loss over the last 16 years within the foraging distribution of the species, with a 20% reduction in forest cover, decreasing from 2 219 170 ha in 2000 to 1 769 488 ha in 2016, while the agropastoral surface increased by 23% (392 916 ha). However, changes over time were not statistically significant (forest surface: Spearman's correlation, $r = -0.38$, $p = 0.13$; agropastoral surface: Spearman's correlation, $r = 0.09$, $p = 0.74$), probably because of high inter-year variability that could be associated with inconsistent measurements or classification of land uses (figure 4).

*Removal of individuals from the wild.* We recorded 103 birds removed from the wild population: 80% of them were poached for illegal trade, 16% were killed by farmers as a consequence of the consumption of corn crops by macaws, 2% died as a result of collisions with power lines and 1% were hunted for indigenous use. The removal of individuals varied over the years (figure 5). The mean number of birds seized after being poached from nests differed between the periods before (1937–2000) and after (2001–2017) the establishment of permanent staff in Jeremoabo (Mann–Whitney $U$-test, $W = 373$, $p = 0.0089$; figure 5). On the other hand, while crop damage by macaws drastically varied across the years (figure 6), there was no clear temporal pattern in the number of macaws killed by farmers (figure 5), and the annual numbers killed were unrelated to the amount of crop damage (Spearman's correlation, $r = -0.129$, $p = 0.68$).

#### 3.1.4.2. Funding allocation for direct protection

From funds devoted to protection, nothing was invested in protecting foraging areas. However, 84.5% of funds (R$1 294 618) were devoted to protecting the nesting/roosting areas (i.e. creation and surveillance of CBS), thus surely contributing to reducing nest poaching, which causes 80% of the losses of macaws from the wild population. On the other hand, 14.6% (R$223 775) of protection funds were devoted to the corn damage reward program and 0.8% (R$12 785) to rescuing injured birds found in the wild, which

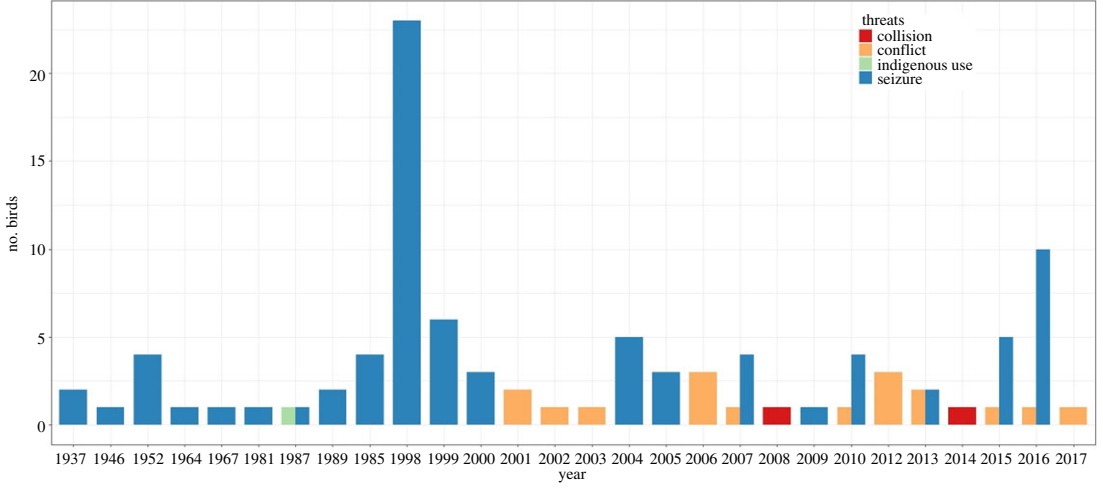

**Figure 5.** Number of Lear's macaws extracted from the wild population over the years as a result of different human-related causes.

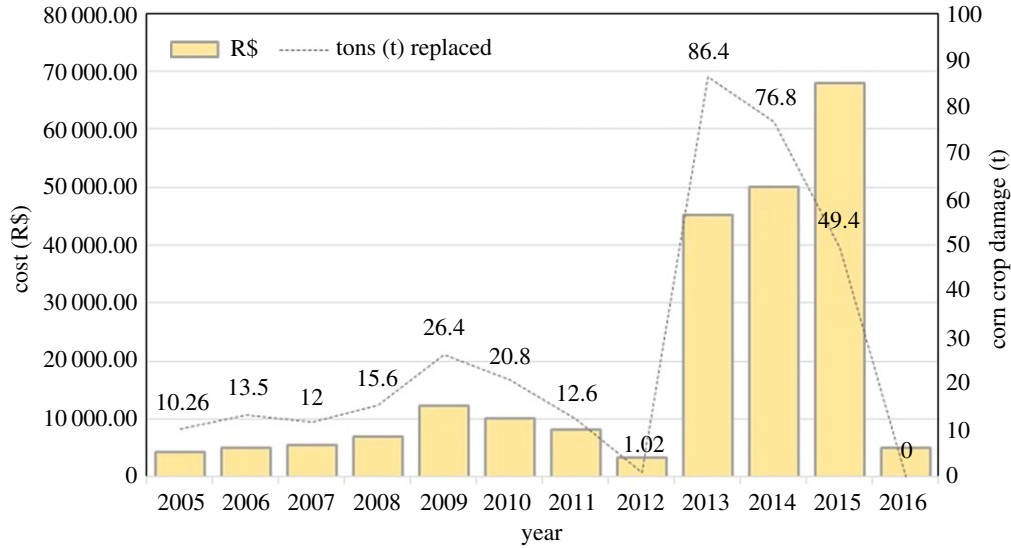

**Figure 6.** Temporal trends in corn crop damage (in tons) and costs to economically compensate the losses to farmers.

caused 16% of the losses of macaws from the wild population. Both indirect measures were aimed at reducing bird mortality due to conflicts with farmers. Finally, no funds were devoted to modifying power lines to reduce collision risks, which causes 2% of the losses of macaws from the wild population. Overall, funds invested to mitigate collisions, the negative effects caused by the human exploitative group (poaching) and by the human damage group (killing) were highly proportional to their effects on removing macaws from the wild population (figure 2).

## 3.2. Rewards

### 3.2.1. Conservation rewards

The global population of the Lear's macaw has increased in recent decades (figure 7) in parallel with conservation funding investment (figure 3a; Spearman's correlation, $r = 0.90$, $p < 0.001$, $n = 16$).

### 3.2.2. Economic rewards

#### 3.2.2.1. Ornithological tourism

The four small local birdwatching companies accumulated visits of approximately 50 persons/year. Birdwatchers usually arrive late in the afternoon and spend a single night in the city (Canudos) to

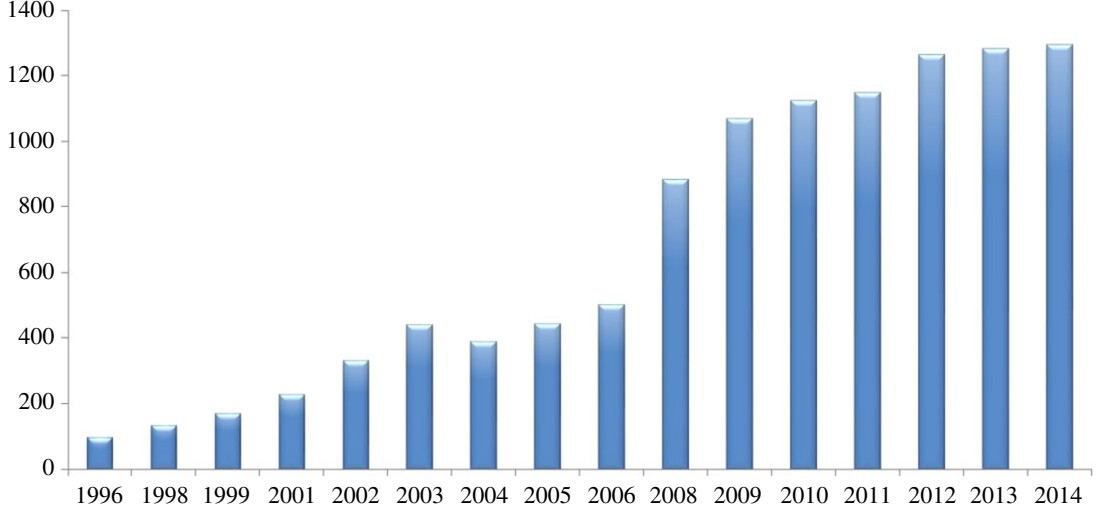

**Figure 7.** Approximate global population trend of the Lear's macaw until the most recent year for which a census is available. Numbers are the mean number of macaws censused each year (figure redrawn from [25]).

take photographs of the macaws at the CBS the following morning, then leave for other destinations. Considering that the average daily accommodation cost in the city is R$40/person (approx. US$12/person) and a meal costs on average R$30 (approx. US$9), ornithological tourism may contribute R$3500/year (approx. US$1050/year) to the local community. This estimate was made assuming all birdwatchers stay at a hotel in the city. This approach is however not realistic, since the CBS itself offers visitors different services, including accommodation and meals. Therefore, most of the direct economic benefits would not be shared with the communities but rather would be directed to the administration of the reserve.

### 3.2.2.2. Handicrafts

The Association of Artisans Lear and Chuquê have 15 and 18 handicraftsmen, respectively. Both associations were structured with funding resources devoted to the conservation of the Lear's macaw. On the other hand, the Association of Artisans from Santa Brígida, consisting of 20 artisans, was responsible for training and the diffusion of production techniques to other communities that joined the Licuri Straw Polo. These three associations accumulated R$55 317 in 2014 in product sales (mean R$18 469; ± 17 497 s.d.), equivalent to approx. US$23 439. Considering the number of craftsmen as a whole [48], the average income of each artisan was R$1044/year (approx. US$442/year), equivalent to R$87 artisan/month (approx. US$37/artisan/month). Taking into account that the average family income of each artisan is R$460/month (approx. US$195/month), the sale of handicrafts contributed 18% of the monthly family income.

## 4. Discussion

### 4.1. Conservation costs: origins and targets

The limited resources dedicated to conservation necessitate the prioritization of strategies for generating greater benefits at a lower cost. However, a long-standing debate about whether the focus should be on ecosystem conservation or on the conservation of species [49–52] confounds managers when it comes to choosing the best strategy. An alternative option is to focus on surrogate species (e.g. umbrella species) with the conservation of the ecosystem as a foundation [51]. Our study focused on estimating conservation costs and rewards, and assessing the level of benefits generated from an approach focused on the recovery of an EN and emblematic species.

Little effort has been made to quantify the conservation costs of a single species [5] and we were unable to find literature on single species conservation costs that detailed funding sources. Most of the studies on conservation expenditure are targeted at taxonomic groups or only analyse one funding source. Therefore, our assessment regarding the resource origins of funding Lear's macaw

conservation can only be compared with a multi-species study conducted by Garnett *et al.* [13]. This study revealed that most of the conservation resources dedicated to Australian EN bird species between 1993 and 2000 were provided by Australian government conservation institutions (86%), the rest being supplied by other state agencies and private initiatives, thus largely agreeing with our results. This is not surprising since previous studies have shown that conservation in Brazil has been mainly funded by the state sector [53,54]. This larger contribution is in agreement with the constitutional obligation of governments and the international commitments established by the Convention on Biological Diversity [48]. However, our analysis of temporal trends in funding origins shows that governmental contribution was delayed in time, with the first conservation actions initiated by international NGOs. The much smaller funding contributions of national NGOs were more evenly distributed over time, while private funds only began in very recent years. The different funding sectors not only contributed differently in their amounts of funds over the years, but also in the targets of their funds. While the government invested more in research, censuses and meetings, the NGOs mostly targeted protection and social activities. This suggests that NGO partners may be more likely to support the most tangible actions.

It is expected that more funds invested in a species would reflect a greater diversity of conservation actions implemented. When assessing where and how resources were distributed towards conservation targets, we found that most of the resources were dedicated to scientific research. Research on EN species is needed for adequate planning of conservation efforts [55]. Research investment in the Lear's macaw is reflected in an increase from six articles published in peer-reviewed journals in 1999 to 17 in 2016 as well as in two master theses and an ongoing PhD thesis. Robust funding amounts were also earmarked for the protection of the species and its breeding habitat, as well as for social activities. These actions responded to the urgent need to save the species from extinction and accounted for a large volume of spending since the acquisition, wages and maintenance of land and property are highly expensive.

Both meeting expenses and monitoring activities (censuses) accounted for only 4% of the total budget. While censuses are needed to determine population trends and the potential effectiveness of the conservation actions applied, meetings allow the exchange of information and participation of different conservation sectors and stakeholders to develop action plans [56,57]. Although only a low proportion (37%) of the actions proposed by the action plan for the Lear's macaw had actually been implemented [58], it is expected that fewer conservation actions would have been applied in its absence.

## 4.2. Mitigation of conservation threats

More than ever, the current loss of biodiversity requires more effective conservation actions. In this regard, we evaluated whether resources were adequately devoted to management strategies aimed at reducing the main threats identified for the species. Our assessment revealed that resources invested to directly or indirectly reduce the removal of individuals from the wild were proportional to the relative impact of the main sources of extraction (poaching, killing by farmers and collision with power lines). In this sense, investment in protecting areas undoubtedly helped to reduce both the poaching and killing of macaws. These resources were concentrated on the breeding and communal roost sites of the CBS, covering only 0.2% (1500 ha) of the area of occurrence of the species (915 066 ha). However, the protection of this main breeding and roosting site, together with the indirect effect of research and monitoring activities that can discourage people from poaching and killing macaws, could have increased the breeding success and survival of adult macaws to the point that it could largely explain the positive population trend of the species (figure 7). Nonetheless, 76% of the area of occurrence of the species is outside any protection regime [25], which includes most of the foraging areas [37,59,60] and two historical breeding sites, one of them recently recolonized by the species [30]. Therefore, some Lear's macaws are still poached, most likely from unprotected nesting sites, to supply the illegal trade, which is primarily international. Moreover, although 24% (221 098 ha) of the Lear's macaw area of occurrence is under some type of protection regime, whether in the form of protected areas or indigenous lands, these habitats still suffer from human impacts [61].

A lower proportion of resources are being devoted to minimizing conflicts with farmers. The project aimed at compensating farmers for the damage to corn crops by Lear's macaw was implemented in 2005. Although Brandt & Machado [27] mentioned that corn was being consumed by Lear's macaws as early as 1968, and warned about possible agricultural conflicts, the first case of macaws killed by farmers was not recorded until 2001. Although we did not find a relationship between the damage caused to corn plantations and the number of macaws killed annually, this relationship could be masked both by the technical difficulties in making accurate damage assessments [62] and by the difficulties in locating

killed macaws. Nonetheless, any reduction in killing pressure may have contributed to the population growth of this long-lived species. Moreover, the rewards programme may contribute not only to conservation, but also to increasing the well-being of people in the local community [63]. In this sense, farmers may have changed their view of the Lear's macaw as a competitor, and the number of killed macaws may have been greater in the absence of the compensatory programme.

Although most of the loss of forest surface area occurred in earlier times [61], our results suggest a further reduction in forested areas in recent decades, coinciding with the implementation of conservation actions for the Lear's macaw. This reduction in foraging areas appears not to have affected the recent recovery of the species, but may limit its further population growth and range expansion to recolonize its historical distribution if deforestation continues. However, no funds were apparently devoted to protecting or restoring foraging habitats, as indicated in the final evaluation report of the Lear's macaw action plan [58], suggesting that these activities are not simple or cheap [9]. In fact, there was an attempt to reforest with licuri palms that rapidly failed because of irrigation problems. In addition to habitat loss, overgrazing by free-ranging cows and goats strongly affects this semi-arid region [61], and, thus, the removal of domestic and invasive fauna would be essential to restoring habitat quality. These management actions present clear social and economic obstacles and require considerable investment, which may explain why no attention has been given to this line of action.

## 4.3. Rewards

### 4.3.1. Conservation rewards

Although increased conservation expenditure generally increases the likelihood that a taxon will recover [64–66], the recovery of a species is not solely determined by the amount of money spent. Evidently, it is likewise important how the funds are spent. Our results showed a strong positive association between the amount of annual funding allocated to a species and its population trends, suggesting that conservation funds effectively contributed to the recovery of the species. Some conservation actions, including research, protection and social activities, received a larger amount of funds. Research provided important information on the ecology of a recently discovered species [27], such as the identification of main food sources and foraging areas [28,29,37–39,59,60], of daily foraging movements [31], of extinct and new nesting and roosting sites [32,36], of past and current threats [25,30] (nest poaching, killing adults for food and to avoid crop predation, collision with power lines, competition with Africanized honeybees for nesting sites, habitat degradation), of factors affecting breeding success and population genetics of the species [30]. This amount of information was important to guide conservation actions. Although no actions were addressed to effectively conserve and restore foraging habitats, those aimed at reducing the loss of chicks and adults (protection of nesting and roosting sites, crop-loss compensation) had undoubtedly contributed to the recovery of the species. In addition, social and research activities may also have ultimately contributed to raising awareness and attracting the attention of the local community to the conservation of the species. We cannot rule out the synergistic effect of these two activities by attenuating the removal pressure on birds. There is the possibility that all these combined actions resulting in a reduction in poaching pressure and mortality of adults were enough to explain most of the positive population growth rate by increasing the breeding success and survival rates of the species without a substantial increase in funds for other actions, and, thus, that the relationship between population recovery and increase in funds with time produced a somewhat spurious correlation. However, there is evidence that conservation actions have helped in the recovery of the species, but presumably those actions depend on investments. In the absence of these, the effects could be much worse [67,68].

It appears that conservation efforts have been fruitful in achieving the main conservation goal, which was to improve the conservation status of the species from CR to EN, at a cost of *ca* US$3.7 million. The question remains whether this high level of investment is reasonable; in other words, what is the cost-effectiveness ratio of funds invested in Lear's macaw conservation? This can be approached by comparing these conservation costs with those invested in 14 bird species that were also downlisted from CR to EN by the IUCN as a result of conservation efforts [5]. The expenses invested and the number of years elapsed until the dowlisting of the Lear's macaw fits well within the 95% CI of a linear regression model, showing a value even slightly smaller than expected (figure 8). However, conservation efforts for several of the species compared in figure 8 also included *ex situ* management, such as captive breeding and reintroduction programmes that may require large investments [69]. Although Lear's macaws are kept in Brazilian and foreign captive breeding centres, we did not

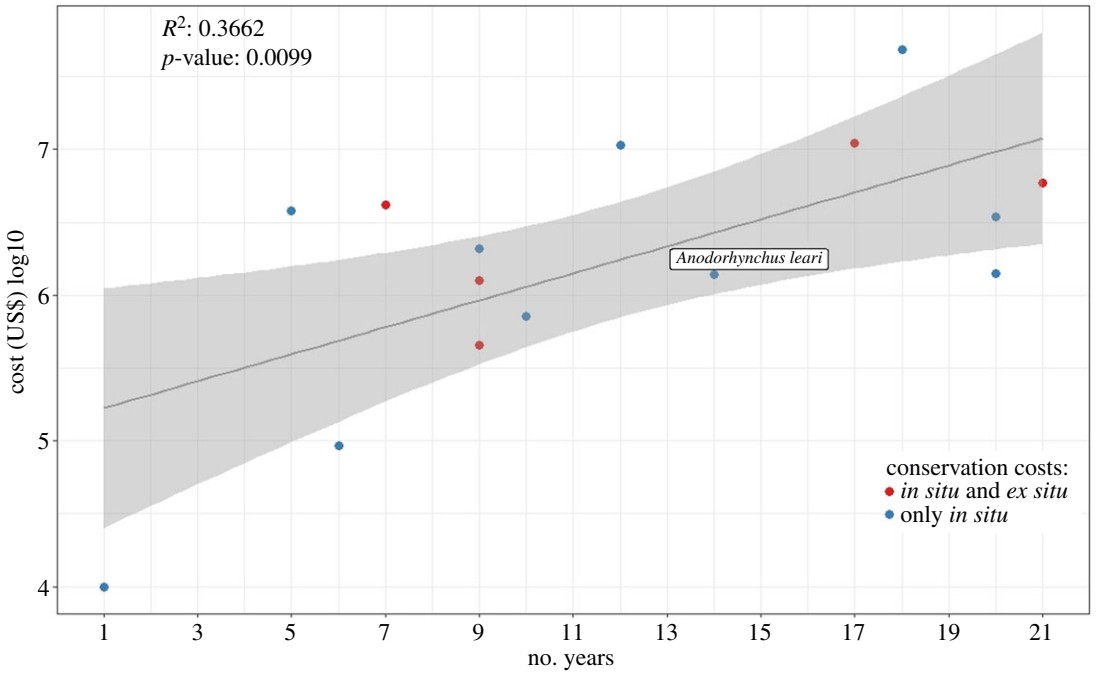

**Figure 8.** Relationship between funds invested in conservation and the number of years until bird species were downlisted from CR to EN, including data from 14 species compiled by McCarthy *et al.* [5] and by this study on the Lear's macaw. Conservation projects that also included *ex situ* actions are differentiated in the figure. Regression line and 95% CI (shaded area) are depicted.

consider *ex situ* management costs since no reintroduction actions were undertaken until 2019, when a first attempt at population reinforcement was attempted, and thus did not contribute to the recovery of the species during the time period examined here. Overall, both the time and funds invested to achieve the goal of recovering the species from its former status as CR seem comparable to other efforts thus far reported worldwide, suggesting an adequate cost-effectiveness ratio.

### 4.3.2. Economic rewards

Birdwatching and income generation projects have been implemented by conservation organizations, governments and development agencies around the world as a tool to achieve conservation and development results, with a generalized promise to generate benefits for conservation and the well-being of the local communities [19,70]. However, quantitative assessments of these activities are scarce and show mixed results in terms of environmental and economic objectives [71–73].

Although some studies indicate that birdwatching [18] and income-generating activities [74] may result in benefits, there is evidence that these activities are of limited impact and are context dependent [71,75]. Our results appear to agree with these assessments, since the economic rewards of the activities undertaken in the area of occurrence of the Lear's macaw were rather anecdotal. The number of birdwatchers visiting the area remains small, and visits are so short that the income generated for local communities is almost nil. An active policy aimed at attracting more birdwatchers—and ecotourists in a broader sense—to view not only this unique macaw species in the wild but also other endemic species of the Caatinga biome [76] and to increase the length of stays, along with a cultural component (e.g. visits to the Canudos War Museum, food tourism, etc.), could substantially increase economic rewards. On the other hand, the contribution of handicrafts to the income of artisans is unquestionable, especially in situations where family income is very low. However, the number of artisans remains small and the benefits of these activities do not seem to be very significant, as they are not able to provide an income equivalent to the minimum wage. Given that handicrafts are mostly reproductions of Lear's macaws and other wildlife using wood and objects made with leaves of the licuri palm (whose nuts are considered the main food source for macaws) (figure 9), visits by birdwatchers and other tourists should be guided to the artisan centres to promote sales. This, along with the promotion of their products in international Brazilian airports and foreign countries through fair trade systems would undoubtedly improve the current situation regarding economic rewards of Lear's macaw conservation.

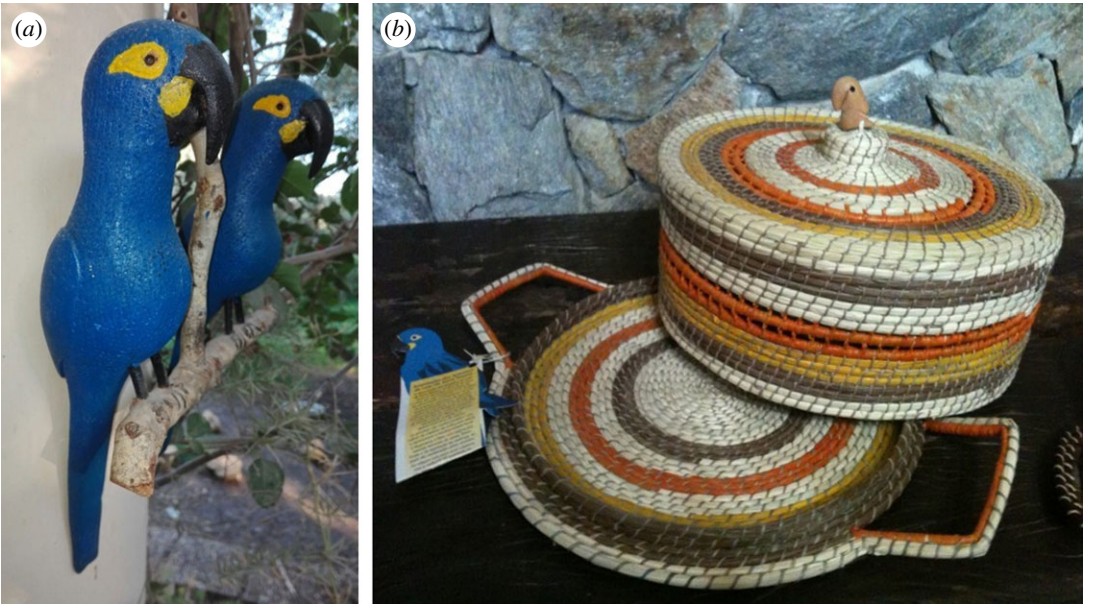

**Figure 9.** Examples of handicrafts related to the conservation of the Lear's macaw made by local artisans using wood (*a*) and leaves of the licuri palm (*b*) (photograph in (*b*) by Simone Tenório).

### 4.3.3. Ecosystem services rewards

Although not measured for our case study, our costs–rewards framework considers ecosystem services rewards. Recent studies show a variety of ecological functions of parrots, including mutualistic interactions, to the extent that they can contribute to shaping the structure and functioning of ecosystems [77–79]. In particular, some macaw species have been shown to be the main seed dispersers of palm trees that are biomass dominant in their ecosystem [77]. This seems also to be the case for the Lear's macaw, an effective disperser of cactus species [80] and several other plant species, including primarily the licuri palm [81]. Therefore, it is expected that the population recovery of the Lear's macaw will contribute to seed dispersal and forest regeneration. The regeneration of licuri palm stands is not only important as a key food source for Lear's macaws and a refuge for many other wildlife species, but also has a social and economic impact, since products obtained from the licuri palm have up to 537 different uses by local communities [26]. Further studies are needed to better identify and quantify the ecosystem services—and derived economic rewards—associated with the recovery of this EN species.

## 5. Conclusion

We present a new costs–rewards framework that integrates detailed information on how and how much different sectors of society contribute to funding the conservation of a threatened species, how funds are distributed among conservation targets and the contributing sectors of society and how these investments drive not only conservation rewards but also economic and ecosystem services. Our results show that the costs involved in the recovery of the Lear's macaw were within those expected to achieve a population increase sufficient to downlist the species from CR to EN, and were proportionally distributed towards the main threats resulting in the removal of individuals from the wild. However, there was a huge discrepancy between the amount invested and the economic benefits that the conservation of this species generates for local communities, which are merely incidental. Additionally, our assessment indicates that we cannot fully assume that conservation actions targeting flagship/umbrella species benefit other species, since no funds were devoted to increasing the extent and quality of natural habitats. Resolving ongoing habitat loss and deterioration by overgrazing is an objective that should be relentlessly pursued, if we want to guarantee the long-term recuperation of this species (still listed as endangered by the IUCN) and an ecosystem that is key for many other endemic wildlife [61].

Data accessibility. Data are available from the Dryad Digital Repository at: https://doi.org/10.5061/dryad.8b5v2r6 [82].
Authors' contributions. A.E.A.B. and J.L.T. designed the work and wrote the manuscript. A.E.A.B. collected and analysed all information. Both authors gave final approval for publication.
Competing interests. The authors declare that they have no competing interests.

Funding. We received no funding for this study.

Acknowledgements. This research would not have been possible without information provided by numerous collaborators. We thank them, especially Nathalia Alves, Dorico Macedo and Ciro Albano, for volunteering their time to help with our cost estimates as well as ICMBio for support and permission to conduct this assessment. We thank M. Moleón and the participants of the Megafauna Workshop at Estación Biológica de Doñana, who inspired us to build the costs–rewards framework presented here, and to two anonymous reviewers for valuable suggestions.

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
