## [Reviewer comments · Royal Society Open Science]

Review History

RSOS-172400.R0 (Original submission)

Review form: Reviewer 1

Is the manuscript scientifically sound in its present form?

No

Are the interpretations and conclusions justified by the results?

No

Is the language acceptable?

Yes

Is it clear how to access all supporting data?

No

Do you have any ethical concerns with this paper?

No

Have you any concerns about statistical analyses in this paper?

No

Recommendation?

Major revision is needed (please make suggestions in comments)

Comments to the Author(s)

See attached word file (Appendix A).

Decision letter (RSOS-172400.R0)

28-Mar-2018

Dear Mr Araujo Barbosa:

Manuscript ID RSOS-172400 entitled "How much does it cost to save a species from extinction? Costs and rewards of conserving the Lear's macaw" which you submitted to Royal Society Open Science, has been reviewed. The comments from reviewers are included at the bottom of this letter.

In view of the criticisms of the reviewers, the manuscript has been rejected in its current form. However, a new manuscript may be submitted which takes into consideration these comments.

Please note that resubmitting your manuscript does not guarantee eventual acceptance, and that your resubmission will be subject to peer review before a decision is made.

Your resubmitted manuscript should be submitted by 25-Sep-2018. If you are unable to submit by this date please contact the Editorial Office.

Please note that Royal Society Open Science will introduce article processing charges for all new submissions received from 1 January 2018. Charges will also apply to papers transferred to Royal Society Open Science from other Royal Society Publishing journals, as well as papers submitted as part of our collaboration with the Royal Society of Chemistry (<http://rsos.royalsocietypublishing.org/chemistry>). If your manuscript is submitted and accepted for publication after 1 Jan 2018, you will be asked to pay the article processing charge, unless you request a waiver and this is approved by Royal Society Publishing. You can find out more about the charges at <http://rsos.royalsocietypublishing.org/page/charges>. Should you have any queries, please contact openscience@royalsociety.org.

Kind regards,
Andrew Dunn

Senior Publishing Editor
Royal Society Open Science
openscience@royalsociety.org

on behalf of Professor Michael Bruford (Associate Editor) and Kevin Padian (Subject Editor)
openscience@royalsociety.org

Editor Comments:

The reviewer feels that the subject of the paper is a worthy one, but has some very serious concerns. On this basis we must recommend a "reject/resub" decision. If the authors choose to resubmit, we will likely send this to the previous reviewer and another one. Thanks and good luck.

Reviewers' Comments to Author:

Reviewer: 1

Comments to the Author(s)

See attached word file

Author's Response to Decision Letter for (RSOS-172400.R0)

See Appendix B.

RSOS-190190.R0

Review form: Reviewer 2

Is the manuscript scientifically sound in its present form?

Yes

Are the interpretations and conclusions justified by the results?

Yes

Is the language acceptable?

Yes

Is it clear how to access all supporting data?

Yes

Do you have any ethical concerns with this paper?

No

Have you any concerns about statistical analyses in this paper?

No

Recommendation?

Accept with minor revision (please list in comments)

Comments to the Author(s)

I very much enjoyed reading this manuscript. And although it provides primarily descriptive statistics on conservation investments and little in the way of analysis or causal inference, I feel it makes a very important contribution to understanding how conservation is funded, and what conservation actions cost in terms of investment. Ideally future work like this can adopt more of an adaptive management approach in which the efficacy of difference conservation actions is measured and also associated costs quantified. This would provide the gold standard for guiding conservation investment decisions.

I appreciate the very detailed accounting of the methods used to acquire financial data, its limitations, and how they handled imperfect information. I think this is especially important since few in conservation would know how to approach this kind of question, and this provides a good methodological roadmap, and reality check, for future work of this kind.

P. 17, l. 15. Population reinforcement: is this captive breeding (and release) costs? If so, please state.

I think the figures could be reduced as there is a fair amount of redundancy. For example, the information in Fig 6 is also in Fig 2 in a different form (% vs \$), Fig 3 could be eliminated in favor of Fig 5 (or, better, provide stacked, color coded columns for annual contributions by different stakeholders), and the details on specific organizations in Fig 4 are not necessary.

Research: is it addressing conservation?

Conservation rewards:

P. 31. The text on conservation application of research needs to be referenced

P. 31, l. 54-6: This statement underscores the major weakness of the study. It necessarily remains very speculative in determining which investments contributed to recovery. That remains unknown and so there is no strong take-away message regarding what worked and what didn't (adaptive management with finances included to generate true cost-effectiveness estimates). Could we have reduced funding for research by 50% an obtained the same result? Did the international NGO investments in social issues pay off?

The papers by Hoffmann et al would be worth reviewing and citing. Although they did not measure funding per sé, they did show that conservation action explained reversals in IUCN status; presumably those actions represented funding.(Hoffmann et al. 2010; Hoffmann et al. 2015)

Hoffmann, M., Duckworth, J.W., Holmes, K., Mallon, D.P., Rodrigues, A.S.L. & Stuart, S.N. (2015) The difference conservation makes to extinction risk of the world's ungulates. *Conservation Biology*, 29, 1303-1313.

Hoffmann, M., Hilton-Taylor, C., Angulo, A., Böhm, M., Brooks, T.M., Butchart, S.H.M., Carpenter, K.E., Chanson, J., Collen, B., Cox, N.A., Darwall, W.R.T., Dulvy, N.K., Harrison, L.R., Katariya, V., Pollock, C.M., Quader, S., Richman, N.I., Rodrigues, A.S.L., Tognelli, M.F., Vié, J.-C., Aguiar, J.M., Allen, D.J., Allen, G.R., Amori, G., Ananjeva, N.B., Andreone, F., Andrew, P., Ortiz, A.L.A., Baillie, J.E.M., Baldi, R., Bell, B.D., Biju, S.D., Bird, J.P., Black-Decima, P., Blanc, J.J., Bolaños, F., Bolivar-G., W., Burfield, I.J., Burton, J.A., Capper, D.R., Castro, F., Catullo, G., Cavanagh, R.D., Channing, A., Chao, N.L., Chenery, A.M., Chiozza, F., Clausnitzer, V., Collar, N.J., Collett, L.C., Collette, B.B., Fernandez, C.F.C., Craig, M.T., Crosby, M.J., Cumberlidge, N.,

Cuttelod, A., Derocher, A.E., Diesmos, A.C., Donaldson, J.S., Duckworth, J.W., Dutson, G., Dutta, S.K., Emslie, R.H., Farjon, A., Fowler, S., Freyhof, J.r., Garshelis, D.L., Gerlach, J., Gower, D.J., Grant, T.D., Hammerson, G.A., Harris, R.B., Heaney, L.R., Hedges, S.B., Hero, J.-M., Hughes, B., Hussain, S.A., Icochea M., J., Inger, R.F., Ishii, N., Iskandar, D.T., Jenkins, R.K.B., Kaneko, Y., Kottelat, M., Kovacs, K.M., Kuzmin, S.L., La Marca, E., Lamoreux, J.F., Lau, M.W.N., Lavilla, E.O., Leus, K., Lewison, R.L., Lichtenstein, G., Livingstone, S.R., Lukoschek, V., Mallon, D.P., McGowan, P.J.K., McIvor, A., Moehlman, P.D., Molur, S., Alonso, A.M., Musick, J.A., Nowell, K., Nussbaum, R.A., Olech, W., Orlov, N.L., Papenfuss, T.J., Parra-Olea, G., Perrin, W.F., Polidoro, B.A., Pourkazemi, M., Racey, P.A., Ragle, J.S., Ram, M., Rathbun, G., Reynolds, R.P., Rhodin, A.G.J., Richards, S.J., Rodríguez, L.O., Ron, S.R., Rondinini, C., Rylands, A.B., Mitcheson, Y.S.d., Sanciangco, J.C., Sanders, K.L., Santos-Barrera, G., Schipper, J., Self-Sullivan, C., Shi, Y., Shoemaker, A., Short, F.T., Sillero-Zubiri, C., Silvano, D.L., Smith, K.G., Smith, A.T., Snoeks, J., Stattersfield, A.J., Symes, A.J., Taber, A.B., Talukdar, B.K., Temple, H.J., Timmins, R., Tobias, J.A., Tsytulina, K., Tweddle, D., Ubeda, C., Valenti, S.V., van Dijk, P.P., Veiga, L.M., Veloso, A., Wege, D.C., Wilkinson, M., Williamson, E.A., Xie, F., Young, B.E., Akçakaya, H.R., Bennun, L., Blackburn, T.M., Boitani, L., Dublin, H.T., Fonseca, G.A.B.d., Gascon, C., Lacher Jr., T.E., Mace, G.M., Mainka, S.A., McNeely, J.A., Mittermeier, R.A., Reid, G.M., Rodriguez, J.P., Rosenberg, A.A., Samways, M.J., Smart, J., Stein, B.A. & N., S.S. (2010) The impact of conservation on the status of the world's vertebrates. *Science*, 330, 1503-1509.

Decision letter (RSOS-190190.R0)

24-Apr-2019

Dear Mr Araujo Barbosa

On behalf of the Editor, I am pleased to inform you that your Manuscript RSOS-190190 entitled "How much does it cost to save a species from extinction? Costs and rewards of conserving the Lear's macaw" has been accepted for publication in Royal Society Open Science subject to minor revision in accordance with the referee suggestions. Please find the referees' comments at the end of this email.

The reviewers and Subject Editor have recommended publication, but also suggest some minor revisions to your manuscript. Therefore, I invite you to respond to the comments and revise your manuscript.

- Ethics statement

- Data accessibility

It is a condition of publication that all supporting data are made available either as supplementary information or preferably in a suitable permanent repository. The data accessibility section should state where the article's supporting data can be accessed. This section should also include details, where possible of where to access other relevant research materials such as statistical tools, protocols, software etc can be accessed. If the data has been deposited in an external repository this section should list the database, accession number and link to the DOI for all data from the article that has been made publicly available. Data sets that have been

deposited in an external repository and have a DOI should also be appropriately cited in the manuscript and included in the reference list.

If you wish to submit your supporting data or code to Dryad (<http://datadryad.org/>), or modify your current submission to dryad, please use the following link:
<http://datadryad.org/submit?journalID=RSOS&manu=RSOS-190190>

- **Competing interests**

- **Authors' contributions**

- **Acknowledgements**

- **Funding statement**

Because the schedule for publication is very tight, it is a condition of publication that you submit the revised version of your manuscript before 03-May-2019. Please note that the revision deadline will expire at 00.00am on this date. If you do not think you will be able to meet this date please let me know immediately.

When submitting your revised manuscript, you will be able to respond to the comments made by the referees and upload a file "Response to Referees" in "Section 6 - File Upload". You can use this

to document any changes you make to the original manuscript. In order to expedite the processing of the revised manuscript, please be as specific as possible in your response to the referees.

on behalf of Professor Michael Bruford (Associate Editor) and Kevin Padian (Subject Editor)
openscience@royalsociety.org

Associate Editor Comments to Author (Professor Michael Bruford):

Associate Editor

Comments to the Author:

Please find some minor amendments required for your MS to become finally accepted. Thanks for submitting this nice paper to the journal.

Reviewer comments to Author:

Reviewer: 2

Comments to the Author(s)

I very much enjoyed reading this manuscript. And although it provides primarily descriptive statistics on conservation investments and little in the way of analysis or causal inference, I feel it

makes a very important contribution to understanding how conservation is funded, and what conservation actions cost in terms of investment. Ideally future work like this can adopt more of an adaptive management approach in which the efficacy of difference conservation actions is measured and also associated costs quantified. This would provide the gold standard for guiding conservation investment decisions.

I appreciate the very detailed accounting of the methods used to acquire financial data, its limitations, and how they handled imperfect information. I think this is especially important since few in conservation would know how to approach this kind of question, and this provides a good methodological roadmap, and reality check, for future work of this kind.

P. 17, l. 15. Population reinforcement: is this captive breeding (and release) costs? If so, please state.

I think the figures could be reduced as there is a fair amount of redundancy. For example, the information in Fig 6 is also in Fig 2 in a different form (% vs \$), Fig 3 could be eliminated in favor of Fig 5 (or, better, provide stacked, color coded columns for annual contributions by different stakeholders), and the details on specific organizations in Fig 4 are not necessary.

Research: is it addressing conservation?

Conservation rewards:

P. 31. The text on conservation application of research needs to be referenced

P. 31, l. 54-6: This statement underscores the major weakness of the study. It necessarily remains very speculative in determining which investments contributed to recovery. That remains unknown and so there is no strong take-away message regarding what worked and what didn't (adaptive management with finances included to generate true cost-effectiveness estimates). Could we have reduced funding for research by 50% and obtained the same result? Did the international NGO investments in social issues pay off?

The papers by Hoffmann et al would be worth reviewing and citing. Although they did not measure funding per sé, they did show that conservation action explained reversals in IUCN status; presumably those actions represented funding. (Hoffmann et al. 2010; Hoffmann et al. 2015)

Hoffmann, M., Duckworth, J.W., Holmes, K., Mallon, D.P., Rodrigues, A.S.L. & Stuart, S.N. (2015) The difference conservation makes to extinction risk of the world's ungulates. *Conservation Biology*, 29, 1303-1313.

Hoffmann, M., Hilton-Taylor, C., Angulo, A., Böhm, M., Brooks, T.M., Butchart, S.H.M., Carpenter, K.E., Chanson, J., Collen, B., Cox, N.A., Darwall, W.R.T., Dulvy, N.K., Harrison, L.R., Katariya, V., Pollock, C.M., Quader, S., Richman, N.I., Rodrigues, A.S.L., Tognelli, M.F., Vié, J.-C., Aguiar, J.M., Allen, D.J., Allen, G.R., Amori, G., Ananjeva, N.B., Andreone, F., Andrew, P., Ortiz, A.L.A., Baillie, J.E.M., Baldi, R., Bell, B.D., Biju, S.D., Bird, J.P., Black-Decima, P., Blanc, J.J., Bolaños, F., Bolivar-G., W., Burfield, I.J., Burton, J.A., Capper, D.R., Castro, F., Catullo, G., Cavanagh, R.D., Channing, A., Chao, N.L., Chenery, A.M., Chiozza, F., Clausnitzer, V., Collar, N.J., Collett, L.C., Collette, B.B., Fernandez, C.F.C., Craig, M.T., Crosby, M.J., Cumberlidge, N., Cuttelod, A., Derocher, A.E., Diesmos, A.C., Donaldson, J.S., Duckworth, J.W., Dutson, G., Dutta, S.K., Emslie, R.H., Farjon, A., Fowler, S., Freyhof, J.r., Garshelis, D.L., Gerlach, J., Gower, D.J., Grant, T.D., Hammerson, G.A., Harris, R.B., Heaney, L.R., Hedges, S.B., Hero, J.-M., Hughes, B., Hussain, S.A., Icochea M., J., Inger, R.F., Ishii, N., Iskandar, D.T., Jenkins, R.K.B., Kaneko, Y., Kottelat, M., Kovacs, K.M., Kuzmin, S.L., La Marca, E., Lamoreux, J.F., Lau, M.W.N., Lavilla, E.O., Leus, K., Lewison, R.L., Lichtenstein, G., Livingstone, S.R., Lukoschek, V., Mallon, D.P.,

McGowan, P.J.K., McIvor, A., Moehlman, P.D., Molur, S., Alonso, A.M., Musick, J.A., Nowell, K., Nussbaum, R.A., Olech, W., Orlov, N.L., Papenfuss, T.J., Parra-Olea, G., Perrin, W.F., Polidoro, B.A., Pourkazemi, M., Racey, P.A., Ragle, J.S., Ram, M., Rathbun, G., Reynolds, R.P., Rhodin, A.G.J., Richards, S.J., Rodríguez, L.O., Ron, S.R., Rondinini, C., Rylands, A.B., Mitcheson, Y.S.d., Sanciangco, J.C., Sanders, K.L., Santos-Barrera, G., Schipper, J., Self-Sullivan, C., Shi, Y., Shoemaker, A., Short, F.T., Sillero-Zubiri, C., Silvano, D.L., Smith, K.G., Smith, A.T., Snoeks, J., Stattersfield, A.J., Symes, A.J., Taber, A.B., Talukdar, B.K., Temple, H.J., Timmins, R., Tobias, J.A., Tsytsulina, K., Tweddle, D., Ubeda, C., Valenti, S.V., van Dijk, P.P., Veiga, L.M., Veloso, A., Wege, D.C., Wilkinson, M., Williamson, E.A., Xie, F., Young, B.E., Akçakaya, H.R., Bennun, L., Blackburn, T.M., Boitani, L., Dublin, H.T., Fonseca, G.A.B.d., Gascon, C., Lacher Jr., T.E., Mace, G.M., Mainka, S.A., McNeely, J.A., Mittermeier, R.A., Reid, G.M., Rodriguez, J.P., Rosenberg, A.A., Samways, M.J., Smart, J., Stein, B.A. & N., S.S. (2010) The impact of conservation on the status of the world's vertebrates. *Science*, 330, 1503-1509.

Author's Response to Decision Letter for (RSOS-190190.R0)

See Appendix C.

Decision letter (RSOS-190190.R1)

04-Jun-2019

Dear Mr Araujo Barbosa,

I am pleased to inform you that your manuscript entitled "How much does it cost to save a species from extinction? Costs and rewards of conserving the Lear's macaw" is now accepted for publication in Royal Society Open Science.

Kind regards,
 Alice Power
 Royal Society Open Science
openscience@royalsociety.org

on behalf of Professor Michael Bruford (Associate Editor) and Kevin Padian (Subject Editor)
openscience@royalsociety.org

Appendix A

To the authors:

This paper takes on the very important task of quantifying the costs incurred in the conservation of the Lear's Macaw in the process of downlisting it from CR to EN. More papers like this are needed so the authors should be applauded for this attempt. The abstract indicates that the paper "developed a costs-rewards framework that integrates information on which sectors of society contribute to funding conservation, how much is contributed, how funds are distributed among conservation targets, and how these investments drive not only conservation rewards but also the economic and ecosystem services that benefit society." The authors do a good and detailed job of providing information on what sectors of society contributed and how each sector's contributions were used. This part is very informative and provides a great deal of insight in to the workings of this large and complex conservation program.

However, the article mostly fail's to achieve the second set of objectives "and how these investments drive not only conservation rewards but also the economic and ecosystem services that benefit society." As a result, this second point becomes a main focus of this review (see Review Table 1, below). Some expenditures were discussed and judged in the absence of much data or support from the literature making these judgments somewhat suspect. Other expenditures (like research, that made up 51% of the budget) were not discussed at all. The absence of a detailed discussion of research is a rather grave error and its omission is highly puzzling as it seems as though it would be relatively simple to provide a fairly detailed evaluation of the importance of research to the recovery of this species.

Review Table 1:

Activity type	Cost data in \$\$	Benefit data \$\$	Benefit data conservation	Benefit summary
Meetings	yes	no	No data, no literature	Presented as inefficient in the absence of data and support from the literature
Protection	Yes	No	Yes	See subsections below
Exploitation at nest/roost sites	Yes	No	Yes (discuss but with little data)	This was likely the MOST important activity done to help the species (according to this reviewer). There is no data driven discussion of this. In addition, the authors seem to waiver or flip-flop on whether or not it was important and how successful it was.
Damage (Corn replacement)	Yes	No	Yes (discuss but with little data)	Presented as inefficient by the authors in the absence of data or literature that support this contention.
Research	Yes	No	No	Not addressed at all. This is the largest and most glaring oversight in this paper.
Census	Yes	No	Yes, clearly needed for	This section was ok. However, no comparison was made with cost of other

			monitoring population	avian censuses to determine if it was cost efficient.
Population reinforcement	Yes	No	Not discussed	Never explained what was done here it is uncertain what it means to have the population “reinforced” (but apparently it was not done using reintroduced birds).
Social	Yes	Not much		There is no indication of how much was spent on different social activities and which of those activities were considered beneficial. The exceptions seem to be handicrafts and birdwatching, but as mentioned below there is no data on the \$\$ spent to promote the individual types of activities.
Handicrafts	No	Yes	No	There is no information on how much conservation funding was invested in promoting this program. The authors considered the program beneficial but somewhat trivial due to low total income for artisans.
Birdwatching	No	Yes	No	There is no information on how much conservation funding was invested in developing or promoting this program. The authors considered the program trivial (and rightfully so) due to low total income
Ecosystem services:	No	No	No (just a brief mention but no real data)	Considered important by the authors, but in the absence of data. The two separate sections dedicated to this topic should be removed from the paper and the topic should just be referred to briefly

See also the comments inserted as notes in the attached PDF file.

Section 3.2.3 and 4.3.3

These sections seem a little too thin to warrant inclusion as a full sections. It is important that parrots are now being shown to disperse seeds, but in the absence of the presentation of any data on this, I think this idea can be confined to the discussion (and perhaps introduction), but it is not needed in the Results, methods, or Abstract. The second author has published a great deal on this topic and is preparing another manuscript on this topic in Lear’s Macaws. In addition, this is an economic analysis paper and there is no estimate of dollar value of this, so it is not warranted as a full section.

Discussion

Page 20 Line 31: Meeting expenses. The authors are clearly biased against holding meetings. Meetings and census BOTH accounted for 4% of the total budget. But the authors chose to present the cost of meetings by stating:

“Meeting expenses were equivalent to resources spent on 10 years of monitoring (censuses) the species, which is needed to determine population trends and the potential effectiveness of the conservation actions applied.”

This clearly makes it seem like meetings are trivial. In a quick review of the two references cited to support their argument that recovery team meetings are not useful, I see no discussion at all about recovery meetings. There is some discussion of the effectiveness of written plans, but not meetings. This section should be eliminated or evaluated in a more evenhanded way. One could counter that the meetings did have many positive impacts that would be hard to quantify without further interviewing of project participants. It is certainly possible that some of the NGOs that participated directly in the meetings may have felt they were an important part of their decisions to keep funding the project. The exchange in information there may have also improved the management of the species in a variety of ways. Neither of these would have been picked up in the current study.

Similarly the part on written reports starting on Page 20 line 42 seems quite thin. The authors cite only a single reference in this section. It states that only 37% of the actions in the Lear’s Action Plan had been implemented. The author imply with this that the plan was a failure. However, this seems like a one-dimensional way of viewing the usefulness of the plan. If there had been no plan, what number of actions would have been implemented? Perhaps quite a few less. In addition, in spite of the fact that only 37% of the activities were implemented, the species populations increased enough to be down listed, again showing the ineffectiveness of percent of actions implemented as a measure of conservation relevance.

On P 20 lines 46 to 52 the authors state that it is hard to evaluate the effectiveness of plans because it is hard to track the actions of managers because managers rarely report their actions. In addition, it is hard to track the costs of the actions taken by the managers. I do not see the logic in this argument. Managers will take actions with or without overall plans. The way to evaluate the efficiency of the plans would be to evaluate whether or not managers carried out more effective actions due to the plans or not. This section should be clarified or removed.

In summary, this entire paragraph should probably be reworked to try and make it a more balanced discussion of meetings and plans, or preferably should be eliminated from the document as there is basically no data on this topic in the paper or in the references cited by the authors.

P 21 L 11. Please remove the word Palliation and derivatives from the entire document. Its definition suggests providing supportive care to terminally ill patients, not finding a cure that enables long-term conservation and survival.

The section 4.2 on Conservation Threats strikes me as somewhat odd. The authors don't seem to communicate the importance and urgency of stopping poaching at the nesting and roosting cliffs. This captured adults in the breeding and non-breeding season and likely disrupted breeding. As a result, this may have eliminated breeding success and removed adults leading to a major reduction in populations. Nowhere do the authors try to communicate the importance of this. As someone familiar with the history of the conservation of the species, the protection of the nesting and roosting grounds were undoubtedly the single most important action taken on behalf of the species. It may have been responsible for the vast majority of the recovery seen to date. Anyone choosing to argue to the contrary would need to provide a detailed argument to the contrary. Also it is important to note that the population HAS recovered to the point of downlisting. However, the authors seem to take an overall negative tone when discussing the protection of the nesting and roosting areas. In particular the authors state that the activities "may have" reduced the number of macaws poached for pet trade. In addition they write a number of other negatively slanted comments like:

1. These facts demonstrate that the efforts devoted to area protection were insufficient to secure the protection of the species within its current range or to cover the spatial expansion of the population.
2. In fact, these investments have helped to alleviate the threats but they did not eliminate them completely. Lear's macaws are still poached, most likely from unprotected nesting sites, to supply the illegal trade.
3. Thus, most poached individuals likely remained undected. In addition, although 24% (221,097.8 ha) of the Lear's macaw occurrence area is under some protection regime, whether in the form of protected areas or indigenous lands, we have no reason to believe that these areas are not subject to strong pressures and impacts (60).

All of these are statements that I would expect in a conservation obituary or call for a last stand, not a species that went from less than 100 to over 1200 in 25 years! The authors should change this tone to highlight the successes and discuss the current shortcomings in management in the light of the overall success.

This also seems to be a good point to bring up an important omission in the analysis of the data. To my knowledge, there are a number of explanatory stories underlying the population data presented in

Figure 10. If I am not mistaken there are publications looking at how discovery of new populations and changes in counting methodology have likely influenced the total counts. I think one paper even tried to say that the increase in counts did not correspond with an increase in population size! I imagine that some of these issues are impacting the data presented in Figure 10. I acknowledge that a detailed discussion of the history of the population counts is beyond the scope of the paper, I think a closer look at these values and a brief history is warranted.

In addition, the authors do not highlight the fact that Figure 10 suggests that the rate of population growth may be slowing down. Up until 2009 the population seems to have been growing faster than it is from 2010 to 2014. This could lend support to the arguments of failings in the current management scheme. P 22 line 54. In this section the authors could discuss this apparent leveling off.

Page 22 line 14: The authors seem to be slanted against the corn replenishment program (except for the last sentence in the paragraph). The section starts by saying "LARGE amounts of money were . . ." The truth is that it represents about 3% of the total expenditure on the species. So selling this as a "large amount of money" seems misleading. Again it seems to be biased towards showing the corn replacement program as inefficient. On P 22 line 22 the authors state "We did not find a relationship between the damages caused to corn plantations and the number of macaws killed annually, thus questioning the effectiveness of this compensatory action." I do not see the logic here. You would expect more macaws killed with more damage in the ABSENCE of a compensation program. If the program truly decoupled the relationship that could easily be taken as a measure of SUCCESS! Of course we have no idea what % of the macaws killed were actually reported, so it is hard to use number of macaws known to be killed as a measure.

With regards to habitat, I know that there was a reforestation project with Licuri that was attempted. It was based on irrigation. A government agency planted Licuri, put in an expensive watering system, after a few months, the watering system broke and the trees all died. This could be looked for and added to show that some money was invested in habitat restoration in foraging areas.

Research: According to the data presented in the paper, about US \$ 1.8 million was spent on research. The vast majority came from the government. Yet all that is discussed about this money is

"Research on endangered species is needed for adequate planning of conservation efforts (54). Research investment in Lear's macaw is reflected in an increase from six articles published in peer-reviewed journals in 1999 to 17 in 2016 as well as in two master theses and an ongoing PhD thesis."

This research output in the form of scientific papers is impressive and shows success by one measure. However, the objective is the conservation of the species. Is there any evidence that research

contributed to the recovery of the species? The authors shed doubt on the effectiveness of protecting nesting sites from poaching and compensating farmers for lost corn (ostensibly to keep them from shooting the macaws), yet assume that US\$ 1.8 Million was good for the species without any analysis whatsoever. This seems odd and makes the reader wonder why. The authors could have looked at the state of published knowledge at the start of the management program and then looked to see what information discovered during funded government research actually was used in actions to aid the conservation of the species. In this way, they could have assessed at some level the impact of research on the conservation of the species.

The manuscript has an ethics statement. I am not sure if that is the same as a US conflict of interest statement. If it is, I think there are a few things that would be included here:

CEMAVE is a government agency that is charged with the conservation of birds in Brazil. If a scientific paper like this were to show that government funds were being spent in a way that was considered “inefficient” that would reflect badly on the government including CEMAVE. As a result, the lead author who works for CEMAVE certainly has a conflict of interest here. Also, it would be good to know how long the lead author has been working on the species (especially how long he has been working on the species from within CEMAVE). If the author has overseen much of the time frame when the money was spent, or was in any way part of the team that allocated where the money went, then there is certainly a conflict of interest. As a result, any discussion of research efficiency could be colored by this reality. Government money was spent on research so responsible government employees may be hesitant to show inefficiency in the use of those moneys.

Similarly, it would be good to know if the second author, or any of the second author’s graduate students are being funded by programs sponsored by the Brazilian Government. If the second author benefits in any way from federal research funds from Brazil, then he too may be hesitant to criticize the efficiency of these research funds as it could jeopardize future funding. More information on conflict of interest could help explain this.

I am a bit concerned about a statement in the discussion:

“Our results showed a strong positive association between the amount of annual funding allocated for a species and its population trends, suggesting that conservation funds effectively contributed to the recovery of the species.”

I imagine that if you stopped the major loss of adults and chicks, you would expect the population to grow following a logistic growth curve. As a result, the growth could easily match what is seen in the graph in the absence of the extra spending. This “null hypothesis” should be considered and then if the

authors wish to make the claim of how additional money helped beyond that, they should need to show additional evidence.

In summary, this is a very interesting paper. It presents very highly relevant and useful data on who paid how much for the conservation of the Lear's Macaw. However, the discussion is incomplete and seems to be curiously biased for or against certain topics.

Appendix B

Dear editor,

We are delighted with the very detailed suggestions provided by the reviewer. On the other hand, we apologize for our delay in preparing a revised version, due to our multiple field work expeditions and unavoidable compromises we had since we received this review. Here we are providing a fully revised version where we take into account all the suggestions provided by the reviewer in his/her letter and the annotated PDF manuscript, as we detail below. All changes done are shown in the manuscript.

To the authors:

This paper takes on the very important task of quantifying the costs incurred in the conservation of the Lear's Macaw in the process of downlisting it from CR to EN. More papers like this are needed so the authors should be applauded for this attempt. The abstract indicates that the paper "developed a costsrewards framework that integrates information on which sectors of society contribute to funding conservation, how much is contributed, how funds are distributed among conservation targets, and how these investments drive not only conservation rewards but also the economic and ecosystem services that benefit society." The authors do a good and detailed job of providing information on what sectors of society contributed and how each sector's contributions were used. This part is very informative and provides a great deal of insight in to the workings of this large and complex conservation program.

RE: We also think the costs-rewards framework we developed could be applied to a number of other threatened species under conservation management, so this is an important point of our paper as our novel framework could be used by many other researchers and wildlife managers.

However, the article mostly fail's to achieve the second set of objectives "and how these investments drive not only conservation rewards but also the economic and ecosystem services that benefit society." As a result, this second point becomes a main focus of this review (see Review Table 1, below). Some expenditures were discussed and judged in the absence of much data or support from the literature making these judgments somewhat suspect. Other expenditures (like research, that made up 51% of the budget) were not discussed at all. The absence of a detailed discussion of research is a rather grave error and its omission is highly puzzling as it seems as though it would be relatively simple to provide a fairly detailed evaluation of the importance of research to the recovery of this species.

RE: These comments and the summarizing table elaborated by the reviewer (see detailed comments below) greatly helped us to realize our flaws in exposing some results and discussing them. We agree there is still little information on ecosystem services provided by Lear's macaws and it is hard to quantify them. Therefore, we have removed ecosystem services from our objectives and just make brief comments in Discussion. We have not removed them from our costs-rewards framework since this conceptual scheme could be applied to other species, and even ecosystem services provided by Lear's macaws could be better analysed in the future. Regarding research, we now discuss better its implications (see more detailed explanations across this response letter).

Section 3.2.3 and 4.3.3

These sections seem a little too thin to warrant inclusion as full sections. It is important that parrots are now being shown to disperse seeds, but in the absence of the presentation of any data on this, I think this idea can be confined to the discussion (and perhaps introduction), but it is not needed in the Results, methods, or Abstract. The second author has published a great deal on this topic and is preparing another manuscript on this topic in Lear's Macaws. In addition, this is an economic analysis paper and there is no estimate of dollar value of this, so it is not warranted as a full section.

RE: Accordingly, we have removed the corresponding section of Results (no section about ecosystem services was provided in Methods), and have reworded the Discussion section making it shorter. In Abstract, we only mention that ecosystem services have yet to be quantified.

Discussion

Page 20 Line 31: Meeting expenses. The authors are clearly biased against holding meetings. Meetings and census BOTH accounted for 4% of the total budget. But the authors chose to present the cost of meetings by stating:

"Meeting expenses were equivalent to resources spent on 10 years of monitoring (censuses) the species, which is needed to determine population trends and the potential effectiveness of the conservation actions applied."

This clearly makes it seem like meetings are trivial. In a quick review of the two references cited to support their argument that recovery team meetings are not useful, I see no discussion at all about recovery meetings. There is some discussion of the effectiveness of written plans, but not meetings. This section should be eliminated or evaluated in a more evenhanded way. One could counter that the meetings did have many positive impacts that would be hard to quantify without further interviewing of project participants. It is certainly possible that some of the NGOs that participated directly in the meetings may have felt they were an important part of their decisions to keep funding the project. The exchange in information there may have also improved the management of the species in a variety of ways. Neither of these would have been picked up in the current study.

Similarly the part on written reports starting on Page 20 line 42 seems quite thin. The authors cite only a single reference in this section. It states that only 37% of the actions in the Lear's Action Plan had been implemented. The author imply with this that the plan was a failure. However, this seems like a onedimensional way of viewing the usefulness of the plan. If there had been no plan, what number of actions would have been implemented? Perhaps quite a few less. In addition, in spite of the fact that only 37% of the activities were implemented, the species populations increased enough to be down listed, again showing the ineffectiveness of percent of actions implemented as a measure of conservation relevance.

On P 20 lines 46 to 52 the authors state that it is hard to evaluate the effectiveness of plans because it is hard to track the actions of managers because managers rarely report their

actions. In addition, it is hard to track the costs of the actions taken by the managers. I do not see the logic in this argument. Managers will take actions with or without overall plans. The way to evaluate the efficiency of the plans would be to evaluate whether or not managers carried out more effective actions due to the plans or not. This section should be clarified or removed.

In summary, this entire paragraph should probably be reworked to try and make it a more balanced discussion of meetings and plans, or preferably should be eliminated from the document as there is basically no data on this topic in the paper or in the references cited by the authors.

RE: We have reworded and largely shorten this paragraph following these suggestions rather than removing it, as we feel some mention to these expenses is needed in Discussion.

P 21 L 11. Please remove the word Palliation and derivatives from the entire document. Its definition suggests providing supportive care to terminally ill patients, not finding a cure that enables long-term conservation and survival.

RE: Done. We use now words such as mitigation or reducing

The section 4.2 on Conservation Threats strikes me as somewhat odd. The authors don't seem to communicate the importance and urgency of stopping poaching at the nesting and roosting cliffs. This captured adults in the breeding and non-breeding season and likely disrupted breeding. As a result, this may have eliminated breeding success and removed adults leading to a major reduction in populations. Nowhere do the authors try to communicate the importance of this. As someone familiar with the history of the conservation of the species, the protection of the nesting and roosting grounds were undoubtedly the single most important action taken on behalf of the species. It may have been responsible for the vast majority of the recovery seen to date. Anyone choosing to argue to the contrary would need to provide a detailed argument to the contrary. Also it is important to note that the population HAS recovered to the point of downlisting. However, the authors seem to take an overall negative tone when discussing the protection of the nesting and roosting areas. In particular the authors state that the activities "may have" reduced the number of macaws poached for pet trade. In addition they write a number of other negatively slanted comments like:

1. These facts demonstrate that the efforts devoted to area protection were insufficient to secure the protection of the species within its current range or to cover the spatial expansion of the population. 2. In fact, these investments have helped to alleviate the threats but they did not eliminate them completely. Lear's macaws are still poached, most likely from unprotected nesting sites, to supply the illegal trade. 3. Thus, most poached individuals likely remained undected. In addition, although 24% (221,097.8 ha) of the Lear's macaw occurrence area is under some protection regime, whether in the form of protected areas or indigenous lands, we have no reason to believe that these areas are not subject to strong pressures and impacts (60).

All of these are statements that I would expect in a conservation obituary or call for a last stand, not a species that went from less than 100 to over 1200 in 25 years! The authors should change this tone to highlight the successes and discuss the current shortcomings in management in the light of the overall success.

RE: We agree we -unintentionally- gave a negative tone here. Accordingly, we have reworded and shortened the whole paragraph following the reviewer's suggestions.

This also seems to be a good point to bring up an important omission in the analysis of the data. To my knowledge, there are a number of explanatory stories underlying the population data presented in Figure 10. If I am not mistaken there are publications looking at how discovery of new populations and changes in counting methodology have likely influenced the total counts. I think one paper even tried to say that the increase in counts did not correspond with an increase in population size! I imagine that some of these issues are impacting the data presented in Figure 10. I acknowledge that a detailed discussion of the history of the population counts is beyond the scope of the paper, I think a closer look at these values and a brief history is warranted.

In addition, the authors do not highlight the fact that Figure 10 suggests that the rate of population growth may be slowing down. Up until 2009 the population seems to have been growing faster than it is from 2010 to 2014. This could lend support to the arguments of failings in the current management scheme. P 22 line 54. In this section the authors could discuss this apparent leveling off.

RE: To our knowledge, there is not a published paper showing a reanalysis of population trends, but just some preliminary results shown in conferences. Erica Pacifico, within her PhD supervised by the second author of this paper, is modelling population trends taking into account variations among years in census efforts, imperfect detection and the recent discovery of new, small populations. So far, preliminary results suggest that the population growth was even larger than previously thought, as well as the total population size, in recent years. However, this work is still in progress and does not change the overall positive population trend of the species that allowed to downlist it from CR to EN. Therefore, we prefer to maintain the yet published population trend (Fig. 10) and do not discuss it, as it will be discussed in detail in another publication once E. Pacifico will finish her work.

Page 22 line 14: The authors seem to be slanted against the corn replenishment program (except for the last sentence in the paragraph). The section starts by saying "LARGE amounts of money were . . ." "The truth is that it represents about 3% of the total expenditure on the species. So selling this as a "large amount of money" seems misleading. Again it seems to be biased towards showing the corn replacement program as inefficient. On P 22 line 22 the authors state "We did not find a relationship between the damages caused to corn plantations and the number of macaws killed annually, thus questioning the effectiveness of this compensatory action." I do not see the logic here. You would expect more macaws killed with more damage in the ABSENCE of a compensation program. If the program truly decoupled the relationship that could easily be taken as a measure of SUCCESS! Of course we have no idea what % of the macaws killed were actually reported, so it is hard to use number of macaws known to be killed as a measure.

RE: Once again, it seems we unintentionally gave a negative tone in this section of Discussion. We have reworded this paragraph to avoid it.

With regards to habitat, I know that there was a reforestation project with Licuri that was attempted. It was based on irrigation. A government agency planted Licuri, put in an expensive watering system, after a few months, the watering system broke and the trees all died. This could be looked for and added to show that some money was invested in habitat restoration in foraging areas.

RE: We thank the reviewer for this suggestion, we now briefly comment this failed action.

Research: According to the data presented in the paper, about US \$ 1.8 million was spent on research. The vast majority came from the government. Yet all that is discussed about this money is

“Research on endangered species is needed for adequate planning of conservation efforts (54). Research investment in Lear’s macaw is reflected in an increase from six articles published in peer-reviewed journals in 1999 to 17 in 2016 as well as in two master theses and an ongoing PhD thesis.”

This research output in the form of scientific papers is impressive and shows success by one measure. However, the objective is the conservation of the species. Is there any evidence that research contributed to the recovery of the species? The authors shed doubt on the effectiveness of protecting nesting sites from poaching and compensating farmers for lost corn (ostensibly to keep them from shooting the macaws), yet assume that US\$ 1.8 Million was good for the species without any analysis whatsoever. This seems odd and makes the reader wonder why. The authors could have looked at the state of published knowledge at the start of the management program and then looked to see what information discovered during funded government research actually was used in actions to aid the conservation of the species. In this way, they could have assessed at some level the impact of research on the conservation of the species.

RE: We agree with the reviewer we overlooked potential conservation rewards of research. We have added several sentences in the section 4.3.1 (Conservation rewards, page 24, first paragraph) showing the several ways in which research increased the knowledge on the ecology and conservation threats of the species and how it could have guided conservation actions. Although derived actions dealt just with the protection of the species, they could have been enough to allow its population recovery

The manuscript has an ethics statement. I am not sure if that is the same as a US conflict of interest statement. If it is, I think there are a few things that would have been to be included here:

CEMAVE is a government agency that is charged with the conservation of birds in Brazil. If a scientific paper like this were to show that government funds were being spent in a way that was considered “inefficient” that would reflect badly on the government including CEMAVE.

As a result, the lead author who works for CEMAVE certainly has a conflict of interest here. Also, it would be good to know how long the lead author has been working on the species (especially how long he has been working on the species from within CEMAVE). If the author has overseen much of the time frame when the money was spent, or was in any way part of the team that allocated where the money went, then there is certainly a conflict of interest. As a result, any discussion of research efficiency could be colored by this reality. Government money was spent on research so responsible government employees may be hesitant to show inefficiency in the use of those moneys.

Similarly, it would be good to know if the second author, or any of the second author's graduate students are being funded by programs sponsored by the Brazilian Government. If the second author benefits in any way from federal research funds from Brazil, then he too may be hesitant to criticize the efficiency of these research funds as it could jeopardize future funding. More information on conflict of interest could help explain this.

RE: It is true the first author (AEBA) works in CEMAVE since 11 years ago. He participated in the elaboration of the Action Plan for the Lear's macaw and was the coordinator of the Action Plan (second edition) and of the census of the species until 2014. This paper results from his Master Thesis, supervised by the second author (JLT), and in no way the results and discussion of this paper may compromise his position in CEMAVE. On the other hand, JLT has also supervised a PhD (by E. Pacifico) on the population ecology and genetics of the species. JLT did not receive funds from the Brazilian government nor is looking for additional research funds to study the species. Frankly, none of the authors have ethic conflicts nor conflicts of interests with the Brazilian government or other funding bodies. We are not sure whether we should state it in this Ethics Section, so we leave the decision to the editor.

We feel the worries from the reviewer came from our failure to better develop a discussion on the rewards of research activities, which we feel we are now discussing adequately. The reviewer suspected we avoided to criticize the investment on research as the Brazilian government (CEMAVE) could be unhappy. However, a clear demonstration of the fact that we did not try to avoid unhappy reactions by CEMAVE is that our previous, inaccurate wording gave a rather negative tone to the usefulness of the meetings and Action Plan that were also sponsored and coordinated by CEMAVE, as the reviewer noted.

I am a bit concerned about a statement in the discussion:

“Our results showed a strong positive association between the amount of annual funding allocated for a species and its population trends, suggesting that conservation funds effectively contributed to the recovery of the species.”

I imagine that if you stopped the major loss of adults and chicks, you would expect the population to grow following a logistic growth curve. As a result, the growth could easily match what is seen in the graph in the absence of the extra spending. This “null hypothesis” should be considered and then if the authors wish to make the claim of how additional money helped beyond that, they should need to show additional evidence.

RE: We fully agree with this possibility, that protection actions resulting in a reduction of bird losses (poaching and killing) could explain by itself most of the population recovery of the

species. We now discuss it twice in Discussion. However, it is difficult to disentangle the effects of protection from others, as they could act simultaneously and synergistically. For example, as we now discuss more clearly, the continuous -and increasing with years- research activities in the nesting and roosting sites may have discouraged people from poaching chicks and killing adults, thus also contributing in an unmeasurable way to increase the growth rate of the population.

In summary, this is a very interesting paper. It presents very highly relevant and useful data on who paid how much for the conservation of the Lear's Macaw. However, the discussion is incomplete and seems to be curiously biased for or against certain topics.

RE: We feel this revised version presents a much more complete and balanced discussion thanks to the useful suggestions and comments provided by the reviewer.

Appendix C

Dear editor,

We are delighted with the impressions and suggestions by the reviewers. Here we are providing a fully revised version where we take into account all minor suggestions provided by the reviewer. All changes done are shown in the manuscript.

To the authors:

P. 17, l. 15. Population reinforcement: is this captive breeding (and release) costs? If so, please state.

RE: Done.

I think the figures could be reduced as there is a fair amount of redundancy. For example, the information in Fig 6 is also in Fig 2 in a different form (% vs \$), Fig 3 could be eliminated in favor of Fig 5 (or, better, provide stacked, color coded columns for annual contributions by different stakeholders), and the details on specific organizations in Fig 4 are not necessary.

RE: We agree there is redundant and unnecessary information. Therefore, we have removed the figure 4. Additionally, we have transferred the figure 6 to supplementary material to share to whom it may concern. We have combined the figures 3 and 5 for the easiest visualization to readers about investments tendency from each sector.

Research: is it addressing conservation?

RE: Yes. Scientific research aimed at generating or updating information on the species to guide conservation actions.

P. 31. The text on conservation application of research needs to be referenced.

RE: Done.

P. 31, l. 54-6: This statement underscores the major weakness of the study. It necessarily remains very speculative in determining which investments contributed to recovery. That remains unknown and so there is no strong take-away message regarding what worked and what didn't (adaptive management with finances included to generate true cost-effectiveness estimates). Could we have reduced funding for research by 50% and obtained the same result? Did the international NGO investments in social issues pay off?

RE: We thank the reviewer for this suggestion, we have reworded this section and now briefly comment on the articles suggested.